# Chromium Removal from Aqueous Solution Using Natural Clinoptilolite

Tonni Agustiono Kurniawan [1,*], Mohd Hafiz Dzarfan Othman [2], Mohd Ridhwan Adam [2], Xue Liang [3], Huihwang Goh [3], Abdelkader Anouzla [4], Mika Sillanpää [5], Ayesha Mohyuddin [6] and Kit Wayne Chew [7]

[1] College of the Environment and Ecology, Xiamen University, Xiamen 361102, China
[2] Advanced Membrane Technology Research Centre (AMTEC), Faculty of Chemical and Energy Engineering, Universiti Teknologi Malaysia (UTM), Skudai 81310, Malaysia
[3] School of Electrical Engineering, Guangxi University, Nanning 530004, China
[4] Department of Process Engineering and Environment, Faculty of Science and technology, University Hassan II of Casablanca, Mohammedia 28806, Morocco
[5] Department of Chemical Engineering, School of Mining, Metallurgy and Chemical Engineering, University of Johannesburg, Doornfontein 2028, South Africa
[6] Department of Chemistry, School of Science, University of Management and Technology, Lahore 54770, Pakistan
[7] School of Chemistry, Chemical Engineering and Biotechnology, Nanyang Technological University (NTU), Singapore 637459, Singapore
* Correspondence: tonni@xmu.edu.cn

**Abstract:** This work investigates the applicability of clinoptilolite, a natural zeolite, as a low-cost adsorbent for removing chromium from aqueous solutions using fixed bed studies. To improve its removal performance for the inorganic pollutant, the adsorbent is pretreated with NaCl to prepare it in the homoionic form of $Na^+$ before undertaking ion exchange with $Cr^{3+}$ in aqueous solution. This work also evaluates if treated effluents could meet the required effluent discharge standard set by legislation for the target pollutant. To sustain its cost-effectiveness for wastewater treatment, the spent adsorbent is regenerated with NaOH. It was found that the clinoptilolite treated with NaCl has a two-times higher Cr adsorption capacity (4.5 mg/g) than the as-received clinoptilolite (2.2 mg/g). Pretreatment of the clinoptilolite with NaCl enabled it to treat more bed volume (BV) (64 BV) at a breakthrough point of 0.5 mg/L of Cr concentration and achieve a longer breakthrough time (1500 min) for the first run, as compared to as-received clinoptilolite (32 BV; 250 min). This suggests that pretreatment of clinoptilolite with NaCl rendered it in the homoionic form of $Na^+$. Although pretreated clinoptilolite could treat the Cr wastewater at an initial concentration of 10 mg/L, its treated effluents were still unable to meet the required Cr limit of less than 0.05 mg/L set by the US Environmental Protection Agency (EPA).

**Keywords:** adsorption; clinoptilolite; ion exchange; low-cost adsorbent; water pollution; zeolite

## 1. Introduction

Water is a fundamental part of life. Without water, there is no life. Within the scope of UN Sustainable Development Goal (SDG) #6 "Clean water and sanitation for all", the development of sustainable water treatment technologies serves as an enabler for water and sanitation equity. As the lack of clean water in different parts of the world results from water pollution caused by refractory pollutants such as inorganic contaminants, water is a deal breaker for accomplishing the SDGs [1]. However, the presence of heavy me-tals in the aquatic environment due to untreated industrial wastewater effluents in water bodies and their potential effects on living organisms has emerged as one of the major environmental concerns worldwide [2]. Water shortage and safety concerns, exacerbated by increasing water demand and water pollution, also represent major challenges in global efforts to contribute to the UN SDGs, while ensuring the provision of clean water as a basic human right for vulnerable communities [3].

To address the demand of our society for "clean water", various technologies have been developed to deal with the shortage of conventional water resources by harvesting it from non-conventional resources, including treated effluents [4]. As water treatment is crucial to a healthy community and a safe environment, wastewater needs to be treated thoroughly so that it does not harm the environment into which it is discharged. Therefore, any water technology has to meet stringent discharge standards for effluents required by environmental legislation [5]. The technology must also be robust to maintain its performance requirements [6]. Other factors such as the characteristics of wastewater, the legal requirements of residual effluent prior to their discharge, treatment performance, plant flexibility and reliability, and long-term environmental impacts need to be taken into account when selecting the most appropriate technology for wastewater treatment [7].

As environmental legislation imposing effluent limits for wastewater discharged from wastewater treatment plants has become increasingly strict, cost-effective water technologies have been in demand in the global market. New approaches need to be examined to supplement existing conventional treatments such as chemical precipitation [8]. The approaches cover avoiding consumption of excessive chemicals and reducing the generation of toxic sludge or secondary waste post-treatment, while simultaneously improving the ability of treated effluent to comply with the requirements of legislation and reducing energy consumption and treatment costs [9].

As traditional treatment alternatives cannot optimize the removal of target contaminants from industrial wastewater, there is a growing need to develop other environmentally sound technologies that could improve their performance for water treatment applications. For this reason, membrane filtration has been developed for the removal of refractory pollutants in wastewater. Unlike other separation technologies, membrane separation has key benefits such as low environmental pollutant emissions, as it represents a physical separation at moderate operating conditions [10]. Despite the ability of membrane filtration to remove target pollutants from wastewater, its limitations are attributed to its costly treatment costs due to massive energy consumption [11]. They are also not cost-effective to treat polluted wastewater due to heavy metals with concentrations over 100 mg/L [12]. Therefore, the search for alternative treatments has intensified in recent years.

Like membrane filtration, a polluted water environment can be restored using low-cost materials based on a physico-chemical process [13]. Through mass transfer, by which a target pollutant is relocated from the liquid phase to the surface of a solid through physico-chemical interactions [14], adsorption has been widely recognized as a novel strategy for treating wastewater laden with inorganic pollutants [15]. Due to its large surface area, adsorption using activated carbon (AC) can eliminate inorganic pollutants such as metals and other refractory pollutants [16]. Although treated effluents can meet the limit of metal effluent, the utilization of AC remains costly for a large-scale application.

The diverse applications of functional materials for adsorbents have recently responded to the need for cost-effective water treatment. Consequently, there is a growing motivation to use non-conventional materials for the removal of inorganic pollutants from polluted water [17]. Natural resources that are locally available in large quantities such as clinoptilolite can be chemically modified and used as low-cost adsorbents [18]. Conversion of the clinoptilolite into functional materials, which can be utilized for water purification, would add to their commercial value and help users minimize waste disposal costs while providing another option to costly AC [19].

The need for sustainable techniques that do not lead to the generation of hazardous by-products has resulted in the practical utilization of clinoptilolite as an adsorbent for environmental remediation. Natural clinoptilolite has gained popularity due to its ion exchange capability [20]. Large deposits of clinoptilolite in Greece and the UK provide industrial users with cost efficiency. This enables them to treat wastewater laden with heavy metals cost-effectively. The market price of clinoptilolite is about USD 0.4 per kg, depending on its quality [21].

Clinoptilolite, a high-silica member of the heulandite group of natural zeolite, is abundantly available in nature. As a crystalline aluminosilicate from natural resources, zeolite has high cation exchange capacities (CEC) with certain metal ions in the solution [22]. The exchange characteristics of clinoptilolite are attributed to the existence of its ne-gatively charged lattice, which is exchangeable with heavy metals [23]. Since alumunium has one less positive charge than silicon, the framework has a net negative charge of one at the site of each alumunium atom and is counterbalanced by the exchangeable cations such as $Na^+$, $K^+$, and $Mg^{2+}$ [24]. The microporosity and high surface area of clinoptilolite make it widely utilized in applications as an ion exchanger, adsorbent, and separation media.

A preliminary study has been undertaken using clinoptilolite as an adsorbent for Cr removal from aqueous solutions using batch modes [25]. Although batch studies are convenient to assess the removal capability of low-cost adsorbents on target adsorbate, they only yield information on the capacity of the media for target metal ions and the rate of metal uptake [26]. Consequently, there is a growing need to perform fixed bed tests using a column prior to scaling up.

To demonstrate its novelty, this work investigates the applicability of clinoptilolite for the treatment of wastewater laden with Cr(VI) based on fixed bed studies. To enhance its treatment performance for the target pollutant, the clinoptilolite was pretreated with NaCl. Chemical pretreatment of clinoptilolite with NaCl was carried out to prepare the adsorbent in the homoionic form of $Na^+$ prior to ion exchange with $Cr^{3+}$ at acidic conditions [27]. This work also evaluates if treated effluents could meet the required discharge standard imposed by legislation [28]. To sustain its cost-effectiveness for wastewater treatment, spent clinoptilolite was regenerated with NaOH [29]. The performance of clinoptilolite in this work for Cr removal is also compared to that of other studies using similar natural materials.

It is expected that contaminated water laden with Cr could be treated cost-effectively with clinoptilolite. This would assist users in minimizing the treatment cost of their wastewater, while meeting the requirement of discharge effluent standards set by local legislation [30].

## 2. Material and Methods

All the chemicals were of analytical grade, supplied by Merck (US), and used without purification. Deionized water was applied to prepare working solutions and reagents.

### 2.1. Cr(VI) Aqueous Solutions

$K_2Cr_2O_7$ was utilized as a source of Cr(VI) in aqueous solutions [31]. To ensure its purity, before being dissolved in deionized water, the chemical was dried in an oven at 100 °C overnight and cooled in a desiccator at ambient temperature. A stock solution of 50 mg/L was obtained by dissolving 0.1414 g of $K_2Cr_2O_7$ in 100 mL of deionized water, while the Cr concentration in working solution was varied by diluting the stock solution [31].

Prior to its use, the pH of the Cr solution was measured using a pH meter. pH adjustment was undertaken using 0.1 M NaOH and/or 0.1 M $H_2SO_4$, which represent strong alkaline and strong acid, respectively. To analyze the remaining Cr concentration in the samples after treatment, about 0.25 g of 1,5-diphenylcarbazide was dissolved in 50 mL of acetone and stored in a brown bottle [32].

### 2.2. Treatment of Clinoptilolite with NaCl

The adsorbent in this study is natural clinoptilolite. Its physical characteristics are listed in Table 1. Prior to experiments, the adsorbent was treated with 2 M NaCl. The suspension was continuously agitated for 24 h using a rotary shaker. The clinoptilolite was separated from the supernatant using GF/C filters, and the liquid was drained [33]. The washing process was repeated to remove excess NaCl from the surface of the clinoptilolite. Finally, the adsorbent was dried for 3 h and stored until required for further use [34]. To understand the change in its morphology before and after its treatment with NaCl,

the clinoptilolite was characterized with TEM (transmission electron microscope), FTIR (Fourier transformation infrared), and XRD (X-ray diffraction).

**Table 1.** Physical properties of clinoptilolite.

| Property | |
|---|---|
| Solid density (g/cm$^3$) | 2.10 |
| Particle size (mm) | 0.68 |
| Packing density (g/cm$^3$) | 2.25 |
| Total surface area (m$^2$/g) | 800 |
| Cation exchange capacity (meq/g) | 2.50 |

### 2.3. Fixed Bed Study

In a fixed-bed study, a glass column, 50 cm in length with 1.0 cm of internal diameter, was packed with a known mass of an adsorbent. At the bottom of the column, a 1 cm layer of glass beads was fitted. The designated column was filled by the adsorbent. Feeding solutions containing Cr(VI) with a concentration of 10 mg/L were then prepared from the stock solution. After adjusting its pH to optimum based on the results of batch studies, the feeding solutions were introduced at the top of the column, and the column was operated with the feeding solutions flowing from top to bottom [35].

The pH of the effluents was monitored hourly to check if there was any change that might take place [36]. A flow rate of 5.0 mL/min was retained with a peristaltic pump. This flow rate might slightly vary between the runs [37]. The effluent samples were periodically collected by a fraction collector and then analyzed for residual Cr concentrations.

The column operations ended after the saturation point was attained, and there was no difference in concentration between influent and effluent ($C_e/C_o = 1$). In this condition, all surface sites of the zeolite were occupied by adsorbed Cr. The column was washed with deionized water to eliminate unadsorbed metal [38].

### 2.4. Column Regeneration

After complete exhaustion ($C_e/C_o = 1$), the column was desorbed by passing regenerants to recover the accumulated Cr on the saturated adsorbent. 0.1 M NaOH solution was used for the regeneration of spent clinoptilolite. Desorption was terminated as soon as the effluent metal concentration was negligible. Afterwards, the column was rinsed with deionized water at the same flow rate of 5 mL/min until the pH of the effluent was equal to its influent of 7.0–7.2 [39]. To quantify the regeneration efficiency (RE) of the spent adsorbent and evaluate its reusability, the following method was applied [40]:

$$(\%RE) = \frac{(A_r)}{(A_0)} \times 100 \tag{1}$$

where $A_0$ and $A_r$ are the adsorption capacities of the adsorbent before and after regeneration, while the % of loss in the Cr adsorption capacity is the fraction of adsorbate that could no longer be adsorbed to that adsorbed during the first cycle was calculated as:

$$(\%LAC) = \frac{(A - I)}{A} \times 100 \tag{2}$$

where $I$ and $A$ are the amount adsorbed in each subsequent cycle and the amount adsorbed during the first cycle, respectively [41].

### 2.5. Chemical Analysis of Cr Concentration

Changes in the Cr(VI) concentrations in the solution after adsorption treatments were calculated colorimetrically based on the Standard Methods [42]. A purple-violet complex

resulted from the reactions between 1.5-diphenylcarbazide and $Cr^{6+}$ in acidic conditions. Absorbance was determined at wavelength ($\lambda$) 540 nm after 10 min of color development. The least detectable concentration based on this method is 0.005 mg/L as Cr(VI) [42]. In acidic solutions, both $HCrO_4^-$ and $Cr_2O_7^{2-}$ anions can be detected.

### 2.6. Statistical Analysis

To ensure the accuracy of the obtained data, all the studies were undertaken in duplicate. The relative standard deviation of Cr removal for the studies was less than 1.0%. When the relative error exceeded this criterion, a third experiment was carried out [43].

## 3. Results and Discussion

### 3.1. Effect of Chemical Pretreatment

Clinoptilolite contains a complement of exchangeable sodium, potassium, magnesium, and calcium ions, with the selectivity of metals as follows: $K^+ > Mg^{2+} > Ca^{2+} > Na^+$ [44]. To prepare the clinoptilolite in the homoionic form of $Na^+$, it was treated with NaCl before adsorption. This treatment was conducted based on the findings of previous studies that $Na^+$ was the most effective exchangeable ion for the ion exchange of heavy metals [45].

After treatment, certain cations such as $K^+$ and $Ca^{2+}$ were strongly held by the clinoptilolite in preference to $Na^+$. Therefore, $Na^+$ is mostly involved in the ion exchange process [46]. The exposure of clinoptilolite to concentrated NaCl led to the production of the Na-rich sample, and this was attributed to the exchange of $Ca^{2+}$. $Mg^{2+}$ and $K^+$ were not exchangeable with other cations, as they were associated with impurities in the sample. For these reasons, the $K^+$, $Ca^{2+}$, and $Mg^{2+}$ contents in the clinoptilolite could be ignored [47].

### 3.2. Characterization of Clinoptilolite before and after Pretreatment

To understand the difference in its morphology before and after its pretreatment with NaCl, the adsorbent was characterized using TEM, FTIR, and XRD. Figure 1 presents the morphology of clinoptilolite based on TEM analysis with scale bars of 1 μm (a), 500 nm (b), and 200 nm (c), respectively. Figure 1c indicates the lamellar-shaped particles of the clinoptilolite after metal adsorption, as compared to the natural clinoptlolite [48].

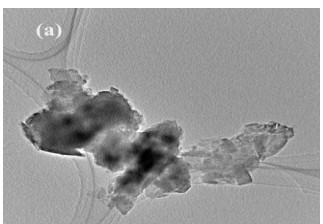 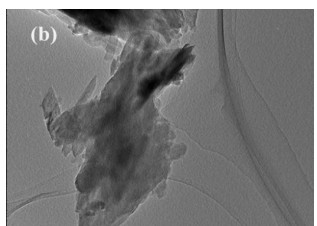 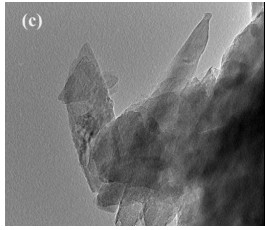

**Figure 1.** TEM characterization of clinoptilolite (**a**) before pretreatment; (**b**) after pretreatment; (**c**) after metal adsorption.

In addition, XRD was used to characterize the crystallinity of the adsorbent (Figure 2). The XRD characterization of the clinoptilolite was indicated by multi-diffraction peaks at 2θ of 9.768°, 11.105°, 13.220°, 16.796°, 18.894°, 20.762°, 22.247°, 22.615°, 25.930°, 26.547°, 28.040°, 29.885°, 31.861°, 32.580°, 36.451°, and 50.051°, as confirmed by the JCPDS card (01-079-1460). The pattern peaks confirmed that clinoptilolite was the main phase of this adsorbent.

To confirm the elemental composition of the clinoptilolite, FTIR was utilized to analyze the functional group on its surface (Figure 3).

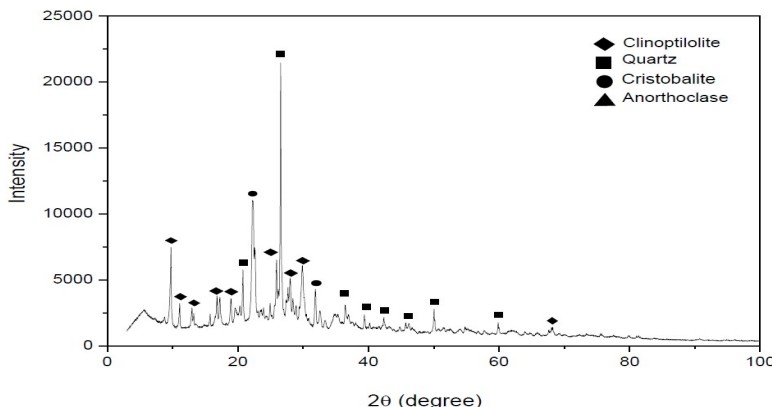

**Figure 2.** XRD analysis of clinoptilolite.

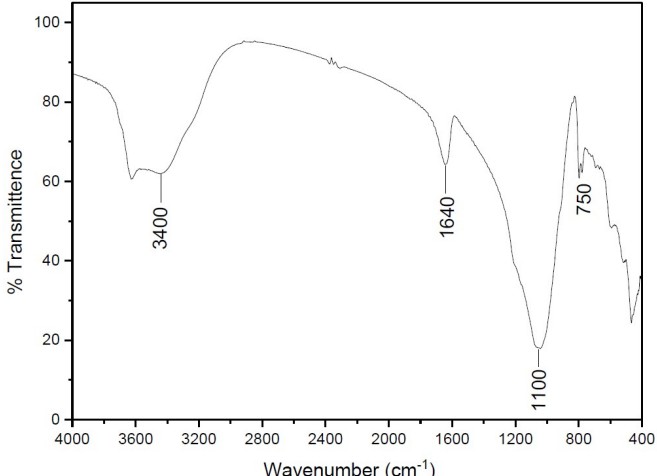

**Figure 3.** FTIR results of clinoptilolite.

Figure 3 shows the existence of spectra related to water molecules at 3400 and 1640 cm$^{-1}$. The asymmetric stretching of the spectrum at 1100 cm$^{-1}$ was associated with the SiO$_4$ tetrahedral [49]. A weaker spectrum at 1000 cm$^{-1}$ was related to the vibration that involves $\equiv$Al$-$O due to the vacancies in Al$^{3+}$. The peak at 750$-$700 cm$^{-1}$ was associated with the symmetric vibration of SiO$_4$, which indicated the presence of Si$-$O$\cdots$HO$-$Si bonds [50].

To further identify the elemental composition of clinoptilolite, X-ray fluorescence (XRF) analysis was carried out. As presented in Table 2, the major elements of the clinoptilolite are SiO$_2$ (73%) ($w/w$), and Al$_2$O$_3$ (12%) ($w/w$). Several trace elements were also present. They included CaO (2.1% ($w/w$), K$_2$O (6.7% ($w/w$)), Fe$_2$O$_3$ (4.3% ($w/w$)), and MgO (1.9% ($w/w$)). Their presence might contribute to Cr adsorption during water treatment [51].

**Table 2.** Composition of clinoptilolite.

| Oxide | Al$_2$O$_3$ | SiO$_2$ | MgO | CaO | Fe$_2$O$_3$ | K$_2$O |
|---|---|---|---|---|---|---|
| Composition (%) | 12.0 | 73.0 | 1.9 | 2.1 | 4.3 | 6.7 |

In adsorption treatment, adsorbate and adsorbent interacted physically in the aqueous phase [52]. As no $\cdot$OH was involved in the degradation of the target pollutant, there was no change in the chemical composition of the starting compounds after treatment [53]. Hence, it is not necessary to prove the stability of the adsorbent before and after treatment [54].

### 3.3. Fixed Bed Studies

In practice, fixed-bed columns are widely used in chemical industries due to their simple and continuous operation [55]. Column operation is essential for the industrial-scale formulation of certain technical systems as it provides credible data on acceptable flow rate, breakthrough time, and loss of adsorption capacity from the first cycle to subsequent cycles [56]. In addition, column studies more accurately quantify the adsorption capacity of an adsorbent for an adsorbate [57]. By using a breakthrough, the practical applicability and feasibility of an adsorbent for Cr removal can be evaluated for industrial application [58].

In column studies, the regeneration of adsorbent and recovery of adsorbate material are the key factors in wastewater treatment applications [59]. To design such an adsorption/desorption process in column operations, the adsorption capacities and adsorption kinetics between adsorbent and adsorbate need to be clearly defined [60]. One way to obtain these characteristics is by examining the concentration of adsorbate in the effluent versus the number of bed volume (BV), which could be treated by adsorbent until reaching complete exhaustion [61]. By using a breakthrough technique, the behavior of metal adsorption on the adsorbent surface can be evaluated [62].

### 3.3.1. Breakthrough

Generally, breakthrough is defined by the point where a specified amount of the influent is detected in the effluent, while the number of bed volume (BV) represents the ratio between the volume of adorbate solution treated and the volume of adsorbent utilized [63]. Both parameters are widely employed to compare the removal performance of adsorbents for certain metal ions [64]. The Cr uptake at the 5% breakthrough point was chosen as the operational capacity of the fixed bed study [65].

To assess its practical utility for Cr removal, column studies were also performed for all types of clinoptilolite. A typical breakthrough curve, representing the ratio of effluent concentration over influent concentration versus the number of BV passed through the column until reaching complete exhaustion, is presented in Figure 4.

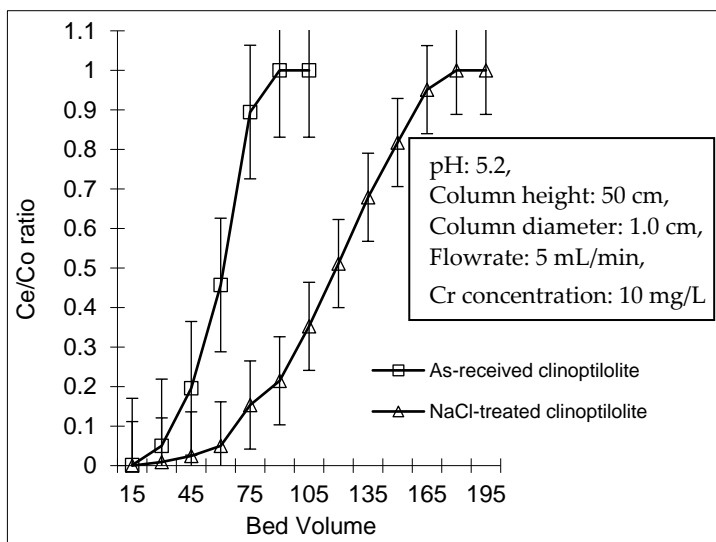

**Figure 4.** Breakthrough curve of as-received clinoptilolite and NaCl-treated clinoptilolite on Cr removal for the first run.

The breakthrough curves of all types of clinoptilolite for the first run, presented in Figure 4, are of the conventional "S" shape. It is interesting to note that the breakthrough point was accomplished when the Cr first appeared in the effluent ($C_e/C_o = 0.05$), while the saturation point was attained in equilibrium when no concentration difference was found between the influent and effluent ($C_e/C_o = 1$) [66].

The two breakthrough curves in the above figure demonstrate that the removal performance of clinoptilolite on Cr was strongly influenced by the way in which the clinoptilolite was treated prior to ion exchange [67]. Figure 4 shows that the complete breakthrough of the as-received clinoptilolite occurred at 32 BV (1.6 L of influent). This indicates that the breakthrough rapidly proceeded at the beginning of the adsorption process, but it started to decline steadily until becoming completely exhausted at 91 BV (5.2 L of feeding solution). At the initial stage of adsorption, when there was an excess of adsorption sites on the clinoptilolite's surface, Cr(III) ions were adsorbed rapidly [68]. However, when the surface sites of clinoptilolite were densely covered by adsorbate, the available sites for metal binding became saturated, resulting in a lower removal rate [69]. It is essential to note that the breakthrough curve of the as-received clinoptilolite in Figure 4 did not follow an ideal "S" shape profile, suggesting the inefficient use of adsorbent and that the large adsorption zone was within the clinoptilolite bed [70].

Compared to the as-received clinoptilolite, the NaCl-treated clinoptilolite achieved the breakthrough point of 0.5 mg/L of effluent Cr concentration remarkably later at 64 BV, corresponding to 3.6 L of influent, and became completely exhausted at 182 BV (about 10.4 L). The significant difference in terms of Cr removal performance between the two types of clinoptilolite suggests that chemical pretreatment of clinoptilolite with NaCl rendered it in the homoionic form of $Na^+$ ($p \leq 0.05$; paired $t$-test). Consequently, the $Na^+$ of the clinoptilolite could be easily replaced by $Cr^{3+}$ in the solution [71]. This provides convincing evidence to explain the proposed adsorption mechanism: that Cr removal by the clinoptilolite resulted from ion exchange, although some might be due to passive physical adsorption [72].

The Cr adsorption capacities of clinoptilolite, determined based on the breakthrough curve area under complete exhaustion, were obtained from the dynamic study by divi-ding the total weight of solute adsorbed by the total weight of adsorbent used. For comparison purposes, the adsorption capacities of clinoptilolite, determined from the batch studies at the same Cr concentration of 10 mg/L, are also presented in Table 3.

**Table 3.** Comparison of the Cr adsorption capacity of clinoptilolite between column and batch studies for the first run.

| Types of Adsorbent | Cr Adsorption Capacity (mg/g) | | Difference of Cr Adsorption Capacity (%) |
|---|---|---|---|
| | Fixed Bed Studies | Batch Studies [25] | |
| As-received clinoptilolite | 2.2 | 1.8 | 23 |
| NaCl-treated clinoptilolite | 4.5 | 3.2 | 41 |

Table 3 shows that the results of column operations were higher than those of batch studies at the same Cr concentration of 10 mg/L. This can be due to the inherent difference in the nature of both studies. In batch studies, the concentration gradient reduced with the longer contact time, while in column studies, the clinoptilolite continuously had phy-sico-chemical interactions with fresh adsorbate solution at the interface of the adsorption zone, as the feeding solution passed through the column [73]. Hence, the concentration gradient increased with a longer residence time. When running columns in series, the first run of column operations needs to be undertaken until attaining complete exhaustion [74].

Monitoring of the effluent pH of NaCl-treated clinoptilolite in column operation shows a substantial increase of pH from 5.0 to 9.0 during the first run; thus, indicating that Cr adsorption on the surface of clinoptilolite releases $OH^-$ into the system [75]. Al-though the column performance of NaCl-treated clinoptilolite tended to deteriorate from the first cycle to the subsequent cycle, it is suggested that NaCl-treated clinoptilolite was a relatively good adsorbent for Cr removal, as this adsorbent exhibited a reasonable Cr adsorption capacity for the first two cycles [76].

3.3.2. Regeneration

Regeneration of a spent adsorbent is necessary when the adsorbent used is expensive or not always available in large quantities [77]. From an economical point of view, an adsorbent can be considered efficient and effective if it is easily regenerated and re-utilized as frequently as possible without altering its removal performance on certain metals [78]. Therefore, column regeneration using a selected chemical was undertaken to restore the removal performance of the same column to its original state [79].

When clinoptilolite in the same column becomes exhausted or when the effluent from an adsorbed bed reaches the allowable discharge level, the recovery of the adsorbed material as well as the regeneration of the adsorbent become necessary [80]. Chemical regeneration is a definite option for this purpose. Therefore, Cr desorption from the clinoptilolite surface was performed with NaOH (Figure 5).

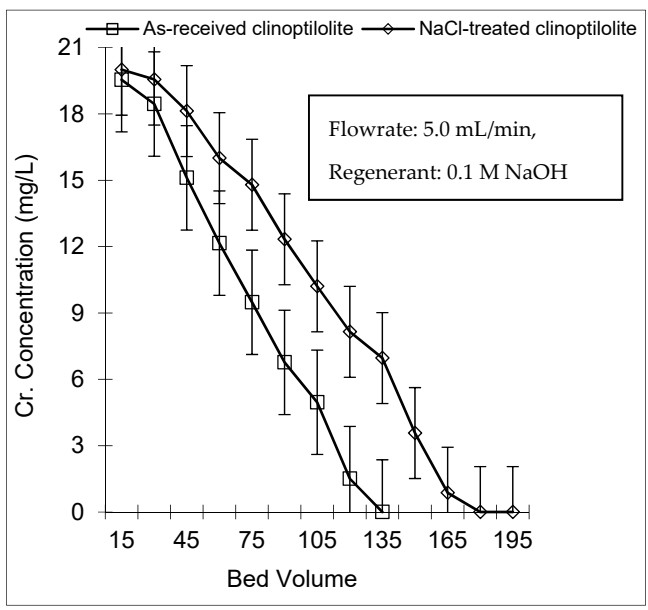

**Figure 5.** Regeneration curve of all types of clinoptilolite by 0.1 M NaOH.

It was found that complete Cr desorption from NaCl-treated clinoptilolite, which occurred at 10.86 L of NaOH (about 164 BV), accounted for the recovery of 93% of Cr from the adsorbent (Table 4); while Cr desorption from the as-received clinoptilolite, which occurred at 96 BV (corresponding to 5.4 L of the same regenerant), provided 91% of Cr recovery. Thus, this suggests that Cr desorption from the former needed an additional volume of regenerant, leading to a costly treatment cost [81].

**Table 4.** Comparison of total mass balance of Cr adsorbed on clinoptilolite before and after regeneration for the first cycle of all types of clinoptilolite.

| Types of Clinoptilolite | Cr before Regeneration (mg/g) | Cr after Regeneration (mg/g) | Regeneration Efficiency (%) * |
|---|---|---|---|
| As-received clinoptilolite | 2.2 | 2.0 | 91 |
| NaCl-treated clinoptilolite | 4.5 | 4.2 | 93 |

Note: * Remarks: % regeneration efficiency (RE) was calculated using Equation (1).

In the first regeneration cycle of the NaCl-treated clinoptilolite and the as-received clinoptilolite, a loss of Cr adsorption capacity of about 23 and 13% was found, respectively (Table 5). This indicates that NaOH is not an ideal regenerant for both types of clinoptilolite,

as their ion exchange capacity with $Cr^{3+}$ tended to decline. Other ion exchangers, such as NaCl, should be tested to desorb Cr from the clinoptilolite's surface, as Cr adsorption on clinoptilolite occurs due to ion exchange between the $Na^+$ of clinoptilolite and the $Cr^{3+}$ in the solution [82]. Thus, this suggests that there is a need to examine the current regeneration procedure more rigorously in order for the regenerated clinoptilolite to adsorb more metals [83].

**Table 5.** Summary of column performance for Cr adsorption by all types of clinoptilolite.

| Type of Clinoptilolite | BV Treated at 1st Run | | Initial Cr Adsorption Capacity (mg/g) | Cr Adsorption Capacity (mg/g) At 2nd Run | Loss of Adsorption Capacity (%) * |
|---|---|---|---|---|---|
| | At Breakthrough | At Exhaustion | | | |
| As-received clinoptilolite | 32 | 91 | 2.2 | 1.7 | 23 |
| NaCl-treated clinoptilolite | 64 | 181 | 4.5 | 3.9 | 13 |

Note: * Remarks: % loss of adsorption capacity was calculated using Equation (2).

Table 5 shows that the Cr adsorption capacity of clinoptilolite remarkably deteriorated over the two cycles. The rate of this deterioration decreased with the increasing number of successive cycles due to exposure to the alkaline regenerant [84]. Although the Cr removal performance of NaCl-treated clinoptilolite in column operation was not excellent, it is important to note that this adsorbent is technically capable of treating Cr-rich effluents at a low cost [85]. Therefore, it needs further consideration before being used for wastewater treatment on an industrial scale.

### 3.3.3. Cr Adsorption Capacity of Clinoptilolite

Clinoptilolite treated with NaCl has a two-times higher Cr adsorption capacity (4.5 mg/g) than as-received clinoptilolite (2.2 mg/g). Pretreatment of clinoptilolite with NaCl enabled it to treat more bed volume (64 BV) at the breakthrough point of 0.5 mg/L of Cr concentration and achieve a longer breakthrough time (1500 min) for the first run, as compared to the as-received clinoptilolite (32 BV; 250 min) because pretreatment of clinoptilolite with NaCl rendered it in the homoionic form of $Na^+$. Consequently, the $Na^+$ of the clinoptilolite could be replaced by $Cr^{3+}$ in the solution via an ion exchange mechanism, and some might be due to passive adsorption [86]. Although the Cr removal performance of NaCl-treated clinoptilolite is not excellent, this adsorbent has a reasonable Cr adsorption capacity (4.5 mg/g). Statistically, the difference in terms of Cr adsorption capacities between treated and untreated clinoptilolite was negligible ($p > 0.05$; paired *t*-test).

To understand the performance of clinoptilolite, its adsorption capacity for Cr in this work was compared to that of previous works for a variety of heavy metals (Table 6). The table shows that the adsorption capacity of clinoptilolite for Cr(III) was comparable to those for Cd(II), Cr(VI), Co(II), Ni(II), Zn(II), and Cu(II). In spite of their low metal adsorption capacities, natural materials, including clinoptilolite, have the ability to remove inorganic pollutants through ion exchange with the target contaminant [87]. It is important to note that the adsorption capacity of an adsorbent varies depending on the initial concentration of adsorbate, the type of adsorbent, and chemical pretreatment [88].

The difference in Cr adsorption capacities between the two chemically treated adsorbents was attributed to the fact that clinoptilolite has fewer negatively charged adsorption sites for Coulombic forces with $Cr^{3+}$. Despite the fact that Cr removal by clinoptilolite occurred due to ion exchange, Cr removal might be due to adsorption on the clinoptilolite surface. Consequently, it resulted in a lower uptake of Cr by clinoptilolite [89].

**Table 6.** An overview of Cr adsorption capacity by different types of zeolite.

| Material | Reference | $Cd^{2+}$ | $Cr^{3+}$ | $Cr^{6+}$ | $Co^{2+}$ | $Ni^{2+}$ | $Zn^{2+}$ | $Cu^{2+}$ | $Pb^{2+}$ |
|---|---|---|---|---|---|---|---|---|---|
| Clinoptilolite | [64] | 2.4 | 0 | | 1.4 | 0.5 | 0.5 | 1.6 | 1.6 |
| | [69] | 1.2 | | | | | | | 1.4 |
| | [62] | 3.7 | | 2.4 | 1.5 | 0.9 | 2.7 | 3.8 | 6.0 |
| | Present study | | | 4.5 | | | | | |
| Chabazite | [70] | 137.0 | | | | | | | 175 |
| | [62] | 6.7 | | 3.6 | 5.8 | 4.5 | 5.5 | 5.1 | 6.0 |
| Chabazite−philipsite | [63] | | 7.1 | | | | | | |
| | [73] | | 0.3 | | | 0.6 | 0.04 | 0.4 | |

Further column studies should be conducted using a flow rate of less than 5 mL/min. Since a low flow rate of feeding solution increases the physico-chemical interaction between clinoptilolite and the target pollutant in the column, the solution has more available residence time to diffuse into the adsorbent for adsorption before it is swept through the column. Hence, it might maximize the treated volume of feeding solutions until breakthrough, extend the lifespan of the bed, and result in a higher Cr removal.

3.3.4. Adsorption Mechanism of Cr Removal by Clinoptilolite

The first step is the reduction of Cr(VI) to Cr(III) (Equation (3)). Although $Cr_2O_7^{2-}$ was utilized as the source of Cr(VI) in aqueous solution, under pH < 6, Cr(VI) exists in the predominant form of $HCrO_4^-$ [90], with the hydrolysis reaction of $Cr_2O_7^{2-}$ as follows:

$$Cr_2O_7^{2-} + H_2O \longleftrightarrow 2\,HCrO_4^- \quad pK_3 = 14.56 \tag{3}$$

The second step controlling Cr removal by clinoptilolite is represented as follows:

$$[Cr(OH)]^{+2}{}_{(s)} + Na_nA_{(z)} + n\,H_2O_{(s)} \longleftrightarrow ([Cr(OH)]^{+2} - H_n - A)_{(z)} + n\,Na^+{}_{(s)} + n\,OH^-{}_{(s)} \tag{4}$$

where A and $n$ represent the adsorption sites on the clinoptilolite's surface and the coefficient of the reaction component, respectively, while subscripts $s$ and $z$ denote the "solution" and "clinoptilolite" phases, respectively. Equation (4) shows that the negative charge of the clinoptilolite, which comes from the tetrahedrally coordinated aluminum, is ba-lanced by the exchangeable $Cr^{3+}$, suggesting that the Cr uptake by the clinoptilolite occurred due to ion exchange and/or adsorption [91]. Cr adsorption is not a fundamentally different process from that of ion exchange. The mechanism of Cr removal by the clinoptilolite in the solution is facilitated by the ion exchange between $Cr^{3+}$ and the $Na^+$ of the clinoptilolite network.

As pH increased to 5.0, the adsorption shifted from left to right, which led to the production of more surface complex ($[Cr(OH)]^{+2}$-$H_n$-A) on the clinoptilolite. The final pH of the solution slightly increased after adsorption because the hydrolysis reaction of the clinoptilolite caused more $OH^-$ release into the solution, resulting in a higher Cr removal. The presence of $OH^-$ in the solution caused $Cr^{3+}$ to be accommodated in the surface lattice of the clinoptilolite, implying that Cr removal by the clinoptilolite is pH-dependent [92].

**4. Conclusions**

This fixed-bed study has revealed the engineering applicability of clinoptilolite as a low-cost adsorbent for treatment of Cr-laden wastewater [85]. The adsorbent pretreated with NaCl had a significantly higher Cr adsorption capacity (4.5 mg/g) as compared to the clinoptilolite in its as-received form (2.2 mg/g). Pretreatment of clinoptilolite with NaCl rendered it in the homoionic form of $Na^+$. Hence, this facilitated the pretreated clinoptilolite to treat more bed volume (64 BV) at the breakthrough point and accomplish a longer time to attain a breakthrough (1500 min) for the first run, as compared to clinoptilolite in its

as-received form (32 BV; 250 min). Despite showing that the pretreated clinoptilolite could treat the Cr-laden wastewater at 10 mg/L of initial Cr concentration, the treated effluents still could not meet the required Cr limit of less than 0.05 mg/L set by the US Environmental Protection Agency (EPA).

**Author Contributions:** Conceptualization, A.A.; methodology, H.G.; validation, X.L. and M.R.A.; formal analysis, K.W.C.; investigation, M.S.; resources, M.H.D.O.; writing—original draft preparation, T.A.K.; writing—review and editing, T.A.K.; supervision, A.M. All authors have read and agreed to the published version of the manuscript.

**Funding:** This work received Research Grants No. Q.J130000.21A6.00P14 and No. Q.J130000.3809. 22H07 from the Universiti Teknologi Malaysia (UTM).

**Data Availability Statement:** Not applicable.

**Conflicts of Interest:** The authors declare no conflict of interest. The funders had no role in the design of the study; in the collection, analyses, or interpretation of data; in the writing of the manuscript, or in the decision to publish the results.

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
