# Peer review of "Chromium Removal from Aqueous Solution Using Natural Clinoptilolite"

_water, doi:10.3390/w15091667_

Round 1
Reviewer 1 Report (Previous Reviewer 1)
The authors of the publication submitted for review have made all the corrections suggested by me and not only by me. The article in its current form has taken on a new dimension. It is correct in terms of content and editorial. In this form, I propose to accept the article for publication.
Thank you for considering my opinion. I encourage the authors to continue working.
Author Response
Manuscript ID: water-2337553
Title: Chromium removal from synthetic wastewater using low-cost clinoptilolite
Reviewer 1
- Comment
The authors of the publication submitted for review have made all the corrections suggested by me and not only by me. The article in its current form has taken on a new dimension. It is correct in terms of content and editorial. In this form, I propose to accept the article for publication..
Reply to Comment:
First of all, the authors are grateful to the reviewer for his/her useful feedbacks to this work. The constructive suggestions have significantly improved the quality of the manuscript after revision based on his/her comments.

Reviewer 2 Report (New Reviewer)
This manuscript reported the adsorption removal of Cr from aqueous solution by the clinoptilolite. The removal of heavy metal ions by solid adsorbent is already not a novel topic. The adsorption property of the clinoptilolite has already been studied. No other new adsorption kinetic and mechanism were proposed in the manuscript. Therefore, I do not recommend this paper to be published in Water.
1) The manuscript is in the marked form, which makes it hard to read.
2) The scale bar of the TEM images in Fig. 1 should be provided.
3) The Cr adsorption capacities of the clinoptilolite with and without NaCl treated was 4.5 mg/g and 2.2 mg/g, which means that the adsorption capacity is very low.
Author Response
Manuscript ID: water-2337553
Title: Cr(VI) removal from contaminated water using low-cost clinoptilolite
First of all, the authors are grateful to the reviewer for his/her useful feedbacks to our work. The constructive suggestions have significantly improved the quality of the manuscript after revision.
Reviewer 2
- Comment
The removal of heavy metal ions by solid adsorbent is not a novel topic. The adsorption property of the clinoptilolite has already been studied.
Reply to Comment:
It is true that the removal of heavy metal ions by solid adsorbent is not novel and that the adsorption by clinoptilolite has already been studied. However, for the authors, novelty means ‘do common things uncommonly”. Previously, metal adsorption by clinoptilolite was traditionally undertaken in batch studies. However, batch study was less convenient to assess the removal capability of low-cost adsorbents on target adsorbate, as it only yields valuable information on the capacity of media for target metal ions and the rate of metal uptake. Consequently, there is a growing need to perform fixed bed tests using a column prior to scaling up.
In practice, fixed-bed column is widely used in chemical industries due to simple and continuous operation [55]. Column operation is essential for industrial scale to formulate certain technical systems, as it provides credible data on acceptable flowrate, breakthrough time, loss of adsorption capacity from the first cycle to subsequent cycles [56]. In addition, column studies more accurately quantify the adsorption capacity of an adsorbent for target adsorbate [57]. By using a breakthrough, the practical applicability and feasibility of adsorbent for Cr removal can be evaluated for industrial application [58]. To demonstrate its novelty, this work investigates the applicability of clinoptilolite for treatment of wastewater laden with Cr(VI) based on fixed bed studies.
Please refer to the additional explanation on page 3 lines 113-119 and on page 6 lines 294-300.
- Comment
No other new adsorption kinetic and mechanism were proposed in the manuscript.
Reply to Comment:
As this work was based on continuous column operations, it was not feasible to determine
the adsorption kinetics of the clinoptolilite on the metal removal. However, the adsorption mechanisms of the clinoptillite on the metal removal has been presented in the manuscript on page 11 lines 464-475.
3.3.4 Adsorption mechanism of Cr removal by clinoptilolite
The first step is through reduction of Cr(VI) to Cr(III) (Equation 3). Although Cr2O72- was utilized as the source of Cr(VI) in aqueous solution, under pH<6, Cr(VI) exists in the predominant form of HCrO4- [78], with the hydrolysis reaction of Cr2O72- as follows:
Cr2O72- + H2O « 2 HCrO4- pK3 = 14.56 (3)
The second step controlling the Cr removal by clinoptilolite is represented as follows:
[Cr(OH)]+2(s)+ NanA(z)+ n H2O(s) ßà ([Cr(OH)]+2-Hn-A)(z)+ n Na+(s)+ n OH-(s) (4)
where: A and n represent the adsorption sites on the clinoptilolite’s surface and the coefficient of reaction component, respectively, while subscripts s and z denote “solution” and “clinoptilolite” phases, respectively.
- Comment
The manuscript is in the marked form, which makes it hard to read.
Reply to Comment:
As suggested, the marked text in the manuscript has been rewritten and revised to make it clear and easily understandable for readers.
- Comment
The scale bar of the TEM images in Fig. 1 should be provided.
Reply to Comment:
As suggested, the scale bars of the TEM images in Figure 1 have been provided on page 5 line 223-224.
Figure 1 presents the morphology of zeolite based on TEM analysis with scale bars of 1μm (a), 500 nm (b), and 200 nm (c), respectively. Figure 1(c) indicated lamellar shaped particle of clinoptilolite after metal adsorption, as compared to raw clinoptlolite.
Figure S1. TEM characterization of zeolite (a) before pretreatment; (b) after pretreatment; (c) after metal adsorption.
- Comment
The Cr adsorption capacities of the clinoptilolite with and without NaCl treated was 4.5 mg/g and 2.2 mg/g, which means that the adsorption capacity is very low.
Reply to Comment:
The additional explanation has been incorporated on page 10 lines 442-448.
Traditionally, metal adsorption capacities of natural materials such as clinoptilolite and chabazite are low. To understand the performance of clinoptilolite, its adsorption capacity for Cr in this work was compared to those of previous works for a variety of heavy metals (Table 6).The table shows that the adsorption capacity of clinoptilolite for Cr(III) was comparable to those for Cd(II), Cr(VI), Co(II), Ni(II), Zn(II), and Cu(II). In spite of their low metal adsorption capacities, natural materials including clinoptilolite have the ability to remove inorganic pollutants through ion exchange with target contaminant [87]. It is important to note that the adsorption capacity of adsorbent vary, depending on the initial concentration of adsorbate, type of adsorbent, and chemical pretreatment [88].
Table 6. An overview of Cr adsorption capacity by different types of clinoptilolite
|
Material |
Reference |
Cd2+ |
Cr3+ |
Cr6+ |
Co2+ |
Ni2+ |
Zn2+ |
Cu2+ |
Pb2+ |
|
Clinoptilolite |
[64] |
2.4 |
0 |
|
1.4 |
0.5 |
0.5 |
1.6 |
1.6 |
|
[69] |
1.2 |
|
|
|
|
|
|
1.4 |
|
|
[62] |
3.7 |
|
2.4 |
1.5 |
0.9 |
2.7 |
3.8 |
6.0 |
|
|
Present study |
|
4.5 |
|
|
|
|
|
|
|
|
Chabazite |
[70] |
137.0 |
|
|
|
|
|
|
175 |
|
[62] |
6.7 |
|
3.6 |
5.8 |
4.5 |
5.5 |
5.1 |
6.0 |
|
|
Chabazite- philipsite |
[63] |
|
7.1 |
|
|
|
|
|
|
|
[73] |
|
0.3 |
|
|
0.6 |
0.04 |
0.4 |
|

Reviewer 3 Report (New Reviewer)
In this manuscript, the results of this research are conveyed thoughtfully and completely, and they are consistent with the experimental findings. However, the authors failed to explain and draw out the novelty of the work, this aspect needs to be improved. This work is worthwhile to be publish in this journal after minor revision. The following issues should be addressed:
1. Introduction is well-organized but the importance and novelty of the research should be highlighted and more clearly stated. The authors should give some examples of works in the bibliography, to clear the advantage of their work in comparison with those works.
2. Maybe the author should compare their results clearly with other reported works, highlighting the advantage and disadvantages of their novel composite.
3. The authors are responsible for the English, which should be polished throughout the manuscript to clear some minor typo/grammar errors.
4. Introduction part, if possible, some important and relative reports references could help:
https://doi.org/10.3390/ma16062170
https://doi.org/10.1007/s10904-023-02604-0
https://doi.org/10.3390/chemengineering7010004
https://doi.org/10.3390/ma15134547
https://doi.org/10.1016/j.ceramint.2022.05.151
5. Stability tests are very important for any material performing as a adsorbent. Any morphological robustness, chemical compositional or oxidation state changes occur after photo-catalysis or not? Need experimental evidences in support of stability.
Hence, I recommend it accepted for publication after minor revisions.
Author Response
Manuscript ID: water-2289752
Title: Chromium removal from synthetic wastewater using low-cost clinoptilolite
First of all, the authors are grateful to the reviewer for his/her useful feedbacks to our work. The constructive suggestions have significantly improved the quality of the manuscript after revision.
Reviewer 3
- Comment
The authors failed to explain and draw out the novelty of the work, this aspect needs to be improved.
Reply to Comment:
The novelty of the work has been highlighted the revised work on page 3 line 112-128.
A preliminary study has been undertaken using clinoptilolite as adsorbent for Cr removal from aqueous solution using batch modes [25]. However, batch study was less convenient to assess the removal capability of low-cost adsorbents on target adsorbate, as it yielded valuable information on the capacity of media for target metal ions and the rate of metal uptake [26]. Consequently, there is a growing need to perform fixed bed tests using a column prior to scaling up.
To demonstrate its novelty, this work investigates the applicability of clinoptilolite for treatment of wastewater laden with Cr(VI) based on fixed bed studies. To enhance its treatment performance for target pollutant, the clinoptilolite was pretreated with NaCl. Chemical pretreatment of clinoptilolite with NaCl was carried out to prepare the adsorbent in the homoionic form of Na+ prior to ion exchange with Cr3+ at acidic conditions [27]. This work also evaluates if treated effluents could meet the required effluent discharge standard imposed by legislation [28]. To sustain its cost-effectiveness for wastewater treatment, spent clinoptilolite was regenerated with NaOH [29]. The performance of clinoptilolite in this work is also compared to those of other studies using similar natural materials. It is expected that contaminated water laden with Cr could be treated cost-effectively with clinoptilolite. This would assist users minimize treatment cost of their wastewater, while meeting the requirement of discharge effluent standard set by local legislation [30].
- Comment
Although introduction is well-organized, the importance and novelty of the research should be highlighted and more clearly stated.
Reply to Comment:
The importance and novelty of the work have been highlighted in the revised text on page 1 lines 43-49 and on page 3 lines 112-125.
The importance:
The presence of heavy metals in the aquatic environment due to untreated industrial wastewater effluents into water body and their potential effects on living organisms has emerged as one of the environmental concerns worldwide [2]. Water shortage and safety concern, exacerbated by the increasing demand and water pollution, also represent major challenges in global efforts to contribute to the SDGs, while ensuring the provision of clean water as basic human right for vulnerable community [3].
The novelty:
A preliminary study has been undertaken using clinoptilolite as adsorbent for Cr removal from aqueous solution using batch modes [25]. However, batch study is less convenient to assess the removal capability of low-cost adsorbents on target adsorbate, as it yields valuable information on the capacity of media for target metal ions and the rate of metal uptake [26]. Consequently, there is a growing need to perform fixed bed tests using a column prior to scaling up.
To demonstrate its novelty, this work investigates the applicability of clinoptilolite for treatment of wastewater laden with Cr(VI) based on fixed bed studies. To enhance its treatment performance for target pollutant, the clinoptilolite was pretreated with NaCl. Chemical pretreatment of clinoptilolite with NaCl was carried out to prepare the adsorbent in the homoionic form of Na+ prior to ion exchange with Cr3+ at acidic conditions [27]. This work also evaluates if treated effluents could meet the required effluent discharge standard imposed by legislation [28]. To sustain its cost-effectiveness for wastewater treatment, spent clinoptilolite was regenerated with NaOH [29]. The performance of clinoptilolite in this work is also compared to those of other studies using similar natural materials.
- Comment
The authors should give some examples of works in the bibliography, to clarify the advantage of their work in comparison with those works.
Reply to Comment:
The additional comparison of their advantages has been incorporated in the revise text. Please refer to the change on page 2 lines 84-106.
Due to its large surface area, adsorption using activated carbon (AC) can eliminate inorganic pollutants such as metals and other refractory pollutants [19]. Although treated effluents can meet the limit of metals effluent, the utilization of AC remains costly for a large-scale application.
The need for sustainable technique, which do not lead to the generation of hazardous by-products, has led to practical utilizations of clinoptilolite as an adsorbent for environmental remediation. Natural zeolite such clinoptilolite has gained popularity due to its ion exchange capability [20]. Large deposits of clinoptilolite in Greece and the UK provide industrial users with cost efficiency. This enables them to treat wastewater laden with heavy metals cost-effectively. The market price of clinoptilolite is about US$ 0.2 per kg, depending on its quality [21].
Clinoptilolite, a high silica member of heulandite group of natural zeolite, is abundantly available in the nature. As a crystalline aluminosilicates from natural resources, zeolite has high cation exchange capacities (CEC) with certain metal ions in the solution [22].
- Comment
The author should compare their results clearly with other reported works, highlighting the advantage and disadvantages of their novel composite.
Reply to Comment:
As requested, Table 6 presents a comparison of present and previous resultsof clinoptilolite for heavy metal removal.
To understand the performance of clinoptilolite, its adsorption capacity for Cr in this work was compared to those of previous works for a variety of heavy metals (Table 6).The table shows that the adsorption capacity of clinoptilolite for Cr(III) was comparable to those for Cd(II), Cr(VI), Co(II), Ni(II), Zn(II), and Cu(II). In spite of their low metal adsorption capacities, natural materials including clinoptilolite have the ability to remove inorganic pollutants through ion exchange with target contaminant. It is important to note that the adsorption capacity of adsorbent vary, depending on the initial concentration of adsorbate, type of adsorbent, and chemical pretreatment [88].
Table 6. An overview of Cr adsorption capacity by different types of clinoptilolite
|
Material |
Reference |
Cd2+ |
Cr3+ |
Cr6+ |
Co2+ |
Ni2+ |
Zn2+ |
Cu2+ |
Pb2+ |
|
Clinoptilolite |
[64] |
2.4 |
0 |
|
1.4 |
0.5 |
0.5 |
1.6 |
1.6 |
|
[69] |
1.2 |
|
|
|
|
|
|
1.4 |
|
|
[62] |
3.7 |
|
2.4 |
1.5 |
0.9 |
2.7 |
3.8 |
6.0 |
|
|
Present study |
|
4.5 |
|
|
|
|
|
|
|
|
Chabazite |
[70] |
137.0 |
|
|
|
|
|
|
175 |
|
[62] |
6.7 |
|
3.6 |
5.8 |
4.5 |
5.5 |
5.1 |
6.0 |
|
|
Chabazite- philipsite |
[63] |
|
7.1 |
|
|
|
|
|
|
|
[73] |
|
0.3 |
|
|
0.6 |
0.04 |
0.4 |
|
5. Comment
The authors are responsible for the English, which should be polished throughout the manuscript to clear some minor typo/grammar errors.
Reply to Comment:
As suggested, a native English speaker has proofread the revised text for its clarity, consistency, logical reasoning, and grammar style.
- Comment
Introduction part, if possible, some important and relative reports references could help:
https://doi.org/10.3390/ma16062170
https://doi.org/10.1007/s10904-023-02604-0
https://doi.org/10.3390/chemengineering7010004
https://doi.org/10.3390/ma15134547
https://doi.org/10.1016/j.ceramint.2022.05.151
Reply to Comment:
As suggested, the following references have been cited and listed in the revised reference list.
Saleh, T.S.; Badawi, A.K.; Salama, R.S.; Mostafa, M.M.M. Design and development of novel composites containing nickel ferrites supported on activated carbon derived from agricultural wastes and its application in water remediation. Materials 2023, 16, 2170. Doi: 10.3390/ ma16062170
Alasri, T.M.; Ali, S.L.; Salama, R.S.; Alshorifi, F.T. Band-structure engineering of TiO2 photocatalyst by AuSe quantum dots for efficient degradation of malachite green and phenol. J. Inorganic Organometallic Polymers and Materials 2023. Doi: 10.1007/s10904-023-02604-0
Baaloudj, O.; Nasrallah, N.; Kenfoud, H.; Bourkeb, K.W.; Badawi, A.K. Polyaniline/ Bi12TiO20 hybrid system for cefixime removal by combining adsorption and photocatalytic degradation. ChemEngineering 2023, 7, 4. Doi: 10.3390/ chemengineering 7010004
Kane, A.; Assadi, A.A.; El Jery, A.; Badawi, A.K.; Kenfoud, H.; Baaloudj, O.; Assadi, A.A. Advanced photocatalytic treatment of wastewater using immobilized titanium dioxide as a photocatalyst in a pilot-scale reactor: Process intensification. Materials 2022, 15, 4547. Doi: 10.3390/ma15134547
Shahzad, W.; Badawi, A. K.; Rehan, Z. A.; Khan, A. M.; Khan, R.A.; Shah, F.; Ali, S.; Ismail, B. Enhanced visible light photocatalytic performance of Sr0. 3 (Ba, Mn) 0.7 ZrO3 perovskites anchored on graphene oxide. Ceramics International 2022, 48, 24979-24988. Doi: 10.1016/ j.ceramint.2022.05.151
- Comment
Stability tests are important for any material performing as an adsorbent. Any morphological robustness, chemical compositional or oxidation state changes occur after photo-catalysis or not? Need experimental evidences in support of stability.
Reply to Comment:
In adsorption treatment, adsorbate and adsorbent interact physically in the aqueous phase. As there is no OH radical involved in the degradation of target pollutant, there is no change in the chemical composition of the starting compounds after treatment. Therefore, it is not necessary to prove the stability of the adsorbent.
Please refer to the explanation on page 6 lines 288-291.

Round 2
Reviewer 2 Report (New Reviewer)
This paper can be accepted in Water.
This manuscript is a resubmission of an earlier submission. The following is a list of the peer review reports and author responses from that submission.
Round 1
Reviewer 1 Report
The article submitted for review describes the research that showed that the chemical pretreatment of zeolite with NaCl improved its Cr adsorption capacities. This topic is known and very often discussed by various research centers. This paper presents research in bed flow reactors and it is a new element. However, I have a few comments:
1. Please explain what 1.79 in line 23 (unit) means?
2. In my opinion, the citation of common information using the latest publications is poor: e.g. citation [1]. This is the case in the introduction. I suggest to refer in the introduction to the issue presented in the title of the article. The first paragraph in the introduction is unnecessary.
3. The conclusion is a repetition of the summary. Please correct it.
4. Please explain why NaOH and H2SO4 were chosen for pH correction?
5. Why are the concentrationsa of solutions sometimes expressed as molar concentrations and sometimes as normal concentrations. Maybe it's worth unifying it.
The publication is a valuable source of information and forms the basis for further analyzes and implementation of new solutions.
Thank you for considering my opinion. I encourage the authors to continue working on improving the manuscript.
Author Response
Manuscript ID: water-2289752
Title: Chromium removal from synthetic wastewater using low-cost clinoptilolite
First of all, the authors are grateful to the reviewer for his/her useful feedbacks to our work. The constructive suggestions have significantly improved the quality of the manuscript after revision.
Reviewer 1
- Comment
Please explain what 1.79 in line 23 (unit) means.
Reply to Comment:
The information presented in the revised manuscript (4.46 and 2.23 mg/g) is the Cr adsorption capacity of the adsorbent based on fixed bed studies. Please refer to the change on page 1 line 28.
- Comment
In my opinion, the citation of common information using the latest publications is poor: e.g. citation [1]. This is the case in the introduction. I suggest to refer in the introduction to the issue presented in the title of the article. The first paragraph in the introduction is unnecessary.
Reply to Comment:
As suggested, the first paragraph has been removed from the revised manuscript. Please refer to the change on page 1 line 38.
- Comment
The conclusion is a repetition of the summary. Please correct it.
Reply to Comment:
As suggested, the conclusion section has been rewritten to reflect the abstract. Please refer to the change on page 9 lines 428-438.
This fixed bed study has revealed the engineering applicability of clinoptilolite as a low-cost adsorbent for treatment of Cr-laden wastewater. The clinoptilolite pretreated with NaCl had significantly higher Cr adsorption capacity (4.46 mg/g), as compared to the clinoptilolite in its as-received form (2.23 mg/g). Pretreatment of clinoptilolite with NaCl rendered it in the homoionic form of Na+. Hence, this facilitated the pretreated clinoptilolite to treat more bed volume (64 BV) at breakthrough point and accomplish a longer time to attain a breakthrough (1600 min) for the first run, as compared to clinoptilolite in its as-received form (32 BV; 300 min). Despite the pretreated clinoptilolite could treat the Cr wastewater at 10 mg/L of initial Cr concentration, the treated effluent still could not comply with the required Cr limit of less than 0.05 mg/L set by the US Environmental Protection Agency (EPA) [75].
- Comment
Please explain why NaOH and H2SO4 were chosen for pH correction?
Reply to Comment:
NaOH and H2SO4 represent strong alkaline and strong acid, respectively. They are commonly used for pH adjustment. Pls refer to the explanation page 3 lines 153-154.
- Comment
Why are the concentrations of solutions sometimes expressed as molar concentrations and sometimes as normal concentration. It is worth to unify it.
Reply to Comment:
As suggested, the unit of the concentration has been unified as ‘molar’. Please refer to the change on page 3 line 153.

Reviewer 2 Report
This manuscript lacks a lot of important data such as the physical and chemical characterization of the zeolite used.
There is a lot of self citation, far beyond the acceptable limit
The manuscript does not present any novel results and the authors have done the same study in 2003, the only difference is that it was done in batch conditions and not fixed bed (Babel, Sandhya, and Tonni Agustiono Kurniawan. "A research study on Cr (VI) removal from contaminated wastewater using natural zeolite." Journal of Ion Exchange 14.Supplement (2003): 289-292.)
- This manuscript is addressing the effectiveness of Cr removal from water via NaCl treated natural zeolite
- the topic is relevant in the field but it is not original and does not address a specific gap in the field because such types of studies are already done before several times
- the manuscript does not add any outstanding or new results to the subject area compared with other published material
- concerning the methodology, the authors should try a new treatment method for the zeolite in order to enhance the Cr removal capacity and compare it with the one they presented in their paper. Moreover, the zeolite must be characterized by XRD, IR, SEM before and after treatment in order to fully understand the its structural characteristics.
- The conclusions are not consistent with the evidence and arguments presented and do they address the main question posed because the researchers have found that the treated effluent was still unable to meet the required Cr limit so what is the importance of this study and how it can be applied. In addition, real water samples must be used in order to further evaluate the removal capacity of this zeolite since real water samples contain ions that may interfere with the removal process thus decreasing Cr removal.
- The references are not appropriate because there is a lot of self-citation, much more the acceptable number
- The presentation of the tables and figures is fine
Author Response
Manuscript ID: water-2289752
Title: Cr(VI) removal from contaminated water using low-cost clinoptilolite
First of all, the authors are grateful to the reviewer for his/her useful feedbacks to our work. The constructive suggestions have significantly improved the quality of the manuscript after revision.
Reviewer 2
- Comment
This manuscript lacks a lot of important data such as the physical and chemical characterization of the zeolite used.
Reply to Comment:
The characterization data of the clinoptilolite in the form of SEM, FTIR, and XRD have been presented in the other manuscript submitted earlier for consideration elsewhere. As it is inappropriate to present the same data sets in the two different work with different publishers, the requested data are not presented in the revised work.
- Comment
There is a lot of self-citation, far beyond the acceptable limit.
Reply to Comment:
Research is a continuous process for every field of study. This work cannot be carried on without referring to previously published works in the area of water treatment using adsorption. Self-citation occurs in this article as the authors obtain and communicate certain ideas effectively from their previous work in this article. Most importantly, self-citation in this work was not conducted to mislead readers.
- Comment
The manuscript does not present any novel results and the authors have done the same study in 2003, the only difference is that it was done in batch conditions and not fixed bed (Babel, Sandhya, and Tonni Agustiono Kurniawan. "A research study on Cr (VI) removal from contaminated wastewater using natural zeolite." Journal of Ion Exchange 14.Supplement (2003): 289-292.).
Reply to Comment:
This work is a series of studies that investigated the applicability of clinoptilolite as an adsorbent for treatment of Cr-laden wastewater using synthetic wastewater at different Cr concentration. To do common things uncommonly and for demonstrating its novelty, fixed bed studies and regeneration tests were undertaken, while comparing the clinoptilolite’s performance in batch and fixed bed studies.

Reviewer 3 Report
This study was systematically conceived and designed to remove Cr from water. However, I have several questions.
First of all, it is not stated which zeolite is used, it is a very broad class of compounds and the zeolite network must be declared, whether it is natural or synthetic, etc. For example. zeolite LTA, its structure collapses in 4M HCl.
Further, synthetic wastewater cannot contain only Cr, adjust the composition to that water that otherwise contains Cr in terms of oxygen content and organic matter, ions (acetate, phosphate...), etc.
Error-values ​​should be included in all measurements. Some kind of material characterisation is necessary, Cr salts can be deposited on the zeolite and this would be seen from the FTIR spectra, so they should be done.
The difference in adsorption of treated and untreated zeolite is negligible. An overview of the obtained results in relation to the values ​​in the literature is not given and adsorption capacities are quite low.
The following sentence is incorrect:
Despite the Cr removal by zeolite mostly occurred due to ion exchange, some Cr removal might be due to adsorption on the zeolite surface also.
The Cr adsorption is not a fundamentally different process from ion exchange. Adsorption occurs precisely because of the electrostatic interactions of Cr with the negative charge of the zeolite network and it is established via ion-ion interactions. What other type of adsorption the authors mean should be clarified.
However, due to the well-designed analysis of the flow system, I suggest a major revision, the authors should correct everything that is requested.
Author Response
Manuscript ID: water-2289752
Title: Chromium removal from synthetic wastewater using low-cost clinoptilolite
First of all, the authors are grateful to the reviewer for his/her useful feedbacks to our work. The constructive suggestions have significantly improved the quality of the manuscript after revision.
Reviewer 3
- Comment
It is not stated which zeolite is used, it is a broad class of compounds and the zeolite network must be declared, whether it is natural or synthetic. For example, zeolite LTA, its structure collapses in 4M HCl.
Reply to Comment:
The adsorbent used in this study is natural zeolite, specifically clinoptilolite. Please refer to the change on page 3 line 158.
- Comment
Synthetic wastewater cannot contain only Cr, adjust the composition to that water that otherwise contains Cr in terms of oxygen content and organic matter, ions (acetate, phosphate...).
Reply to Comment:
Potassium dichromate (K2Cr2O7) was utilized as the source of Cr(VI) in synthetic wastewater. To ensure that only Cr is the composition of the synthetic wastewater, before being dissolved in deionized water, the chemical was dried in an oven at 100oC for 1 h and cooled in a desiccator at ambient temperature. Pease refer to the change on page 3 lines 147-149.
- Comment
Error-values ​​should be included in all measurements.
Reply to Comment:
As requested, error bars have been incorporated in the two figures.
- Comment
Some kind of material characterization is necessary. Cr salts can be deposited on the zeolite and this would be seen from the FTIR spectra.
Reply to Comment:
The characterization data of the clinoptilolite in the form of FTIR have been presented in the other manuscript submitted earlier for consideration elsewhere. As it is inappropriate to present the same data sets in the two different work with different publisher, the requested data are not presented in the revised work.
- Comment
The difference in adsorption of treated and untreated zeolite is negligible. An overview of the obtained results in relation to the values ​​in the literature is not given and adsorption capacities are quite low.
Reply to Comment:
Statistically, the difference in adsorption of treated and untreated clinoptilolite is negligible. As suggested, an overview of the adsorption capacity of clinoptilolite has been provided in Table 2. Please refer to the change on page 9 lines 377-379.
Table 2. An overview of Cr adsorption capacity by different types of zeolite
|
Material |
References |
Cd2+ |
Cr3+ |
Cr6+ |
Co2+ |
Ni2+ |
Zn2+ |
Cu2+ |
Pb2+ |
|
Clinoptilolite |
[64] |
2.40 |
0 |
|
1.42 |
0.48 |
0.50 |
1.64 |
1.60 |
|
[69] |
1.20 |
|
|
|
|
|
|
1.40 |
|
|
[70] |
70.00 |
|
|
|
|
|
|
62.0 |
|
|
[62] |
3.70 |
|
2.40 |
1.50 |
0.90 |
2.70 |
3.80 |
6.00
|
|
|
Present study |
|
4.46 |
|
|
|
|
|
|
|
|
Chabazite |
[70] |
137 |
|
|
|
|
|
|
175 |
|
[62] |
6.70 |
|
3.60 |
5.8 |
4.50 |
5.50 |
5.10 |
6.00 |
|
|
Chabazite- philipsite |
[63] |
|
7.10 |
|
|
|
|
|
|
|
[73] |
|
0.25 |
|
|
0.56 |
0.04 |
0.37 |
|
- Comment
The following sentence is incorrect:
Despite the Cr removal by zeolite mostly occurred due to ion exchange, some Cr removal might be due to adsorption on the zeolite surface also.
Reply to Comment:
Further correction has been provided in the revised text on page 10 lines 413-414.
The Cr uptake by clinoptilolite occurred due to ion exchange and/or adsorption as well [73].
- Comment
The Cr adsorption is not a fundamentally different process from ion exchange. Adsorption occurs because of the electrostatic interactions of Cr with the negative charge of the zeolite network and it is established via ion-ion interactions. What other type of adsorption the authors mean should be clarified.
Reply to Comment:
The required explanation has been presented in the revised text on page 10 lines 414-418.
The Cr adsorption is not a fundamentally different process from ion exchange. Adsorption occurs because of the electrostatic interactions between Cr3+ and the negative charge of the zeolite network and it is established via attractive columbic forces interactions. In addition, the mechanism of Cr removal by zeolite in the solution is facilitated by the ion exchange between Cr3+ and the Na+ of the clinoptilolite network.

Round 2
Reviewer 2 Report
The authors justified the requested comments
Author Response
Manuscript ID: water-2289752
Title: Cr(VI) removal from contaminated water using low-cost clinoptilolite
First of all, the authors are grateful to the reviewer for his/her useful feedbacks to our work. The constructive suggestions have significantly improved the quality of the manuscript after revision.
Reviewer 2
- Comment
The authors justified the requested comments.
Reply to Comment:
The authors are thankful for the reviewer’s understanding.

Reviewer 3 Report
The authors have answered most of the questions raised. Still, there are some unattended issues. There is no simulated wastewater, it is just a solution of targeted ions whose removal is investigated. Remove wastewater from the entire manuscript, it is just a Cr(VI) aqueous solution. The characterization method in other publications is not relevant, especially if it is not yet published. This work still needs a characterisation technique. Please, perform Raman measurements instead.
Author Response
Manuscript ID: water-2289752
Title: Chromium removal from synthetic wastewater using low-cost clinoptilolite
First of all, the authors are grateful to the reviewer for his/her useful feedbacks to our work. The constructive suggestions have significantly improved the quality of the manuscript after revision.
Reviewer 3
- Comment
There is no simulated wastewater. It is just a solution of targeted ions, which removal was investigated.
Reply to Comment:
It is correct that the solution only contains Cr(VI), the target pollutant to be removed. Therefore, the authors used the term “synthetic wastewater” in the previous version to distinguish it from “real wastewater”.
- Comment
Remove the word “wastewater” from the entire manuscript. It is just a “Cr(VI) aqueous solution”.
Reply to Comment:
As requested, the word “wastewater” has been removed throughout the revised text and replaced with “synthetic wastewater” or “Cr(VI) aqueous solution. Please refer to the change on page 1 lines 1, 22, and 146-147.
- Comment
The characterization method in other publications is not relevant, especially if it is not yet published. This work still needs a characterization technique. Please, perform Raman measurements instead.
Reply to Comment:
The characterization data of the work presented in the other journal include FT-IR, FE-SEM, XRD, XPS, TEM, BET, EDS, and Raman spectroscopy. Although the complete data set are not published in the journal, they are resubmitted to the other publisher for re-review purpose. To avoid copyright infringement at later stage, it is safe and necessary to consolidate the complete data set in one place for the benefit of readers.
